# The structure of the COPI coat determined within the cell

Yury S Bykov[1,2], Miroslava Schaffer[3], Svetlana O Dodonova[1†], Sahradha Albert[3], Jürgen M Plitzko[3], Wolfgang Baumeister[3]*, Benjamin D Engel[3]*, John AG Briggs[1,2]*

[1]Structural and Computational Biology Unit, European Molecular Biology Laboratory, Heidelberg, Germany; [2]Structural Studies Division, MRC Laboratory of Molecular Biology, Cambridge, United Kingdom; [3]Department of Molecular Structural Biology, Max Planck Institute of Biochemistry, Martinsried, Germany

**Abstract** COPI-coated vesicles mediate trafficking within the Golgi apparatus and from the Golgi to the endoplasmic reticulum. The structures of membrane protein coats, including COPI, have been extensively studied with *in vitro* reconstitution systems using purified components. Previously we have determined a complete structural model of the *in vitro* reconstituted COPI coat (Dodonova et al., 2017). Here, we applied cryo-focused ion beam milling, cryo-electron tomography and subtomogram averaging to determine the native structure of the COPI coat within vitrified *Chlamydomonas reinhardtii* cells. The native algal structure resembles the *in vitro* mammalian structure, but additionally reveals cargo bound beneath β'–COP. We find that all coat components disassemble simultaneously and relatively rapidly after budding. Structural analysis *in situ*, maintaining Golgi topology, shows that vesicles change their size, membrane thickness, and cargo content as they progress from *cis* to *trans*, but the structure of the coat machinery remains constant.

DOI: https://doi.org/10.7554/eLife.32493.001

*For correspondence:
baumeist@biochem.mpg.de (WB);
engelben@biochem.mpg.de
(BDE);
jbriggs@mrc-lmb.cam.ac.uk
(JAGB)

Present address: †Department
of Molecular Biology, Max Planck
Institute for Biophysical
Chemistry, Göttingen, Germany

Competing interests: The
authors declare that no
competing interests exist.

Reviewing editor: Margaret S.
Robinson, University of
Cambridge, United Kingdom

## Introduction

Vesicular transport between cellular compartments is a fundamental mechanism of eukaryotic cells. Three conserved, archetypal protein coats, COPI, COPII and clathrin, mediate the formation and trafficking of vesicles in the endocytic-secretory pathway (*Bonifacino and Glick, 2004*). These coats share organizational and functional principles, and likely diverged from an ancestral coat prior to the last common eukaryotic ancestor.

The COPI coat mediates intra-Golgi and retrograde Golgi-ER trafficking (*Malhotra et al., 1989*; *Letourneur et al., 1994*), and is essential for maintaining the polarized Golgi structure (*Bonifacino and Glick, 2004*; *Glick and Nakano, 2009*; *Popoff et al., 2011*). COPI is recruited to the membrane as a heptameric complex, called coatomer, by interaction with the small GTPase Arf1. Analogous to the clathrin and COPII coats, COPI coatomer incorporates a coat-like and an adaptor-like subcomplex. The coat-like subcomplex of COPI consists of ε, α, and β' subunits, the latter two of which have a 'proto-coatomer' architecture characterized by N-terminal β–propellers combined with extended α–solenoids. Similar architecture is also found in the clathrin heavy chain, the Sec31 COPII coat subunits, and some nuclear pore components (*Lee and Goldberg, 2010*; *Devos et al., 2004*). The COPI adaptor-like subcomplex consists of γ, ζ, β, and δ subunits and is a homolog of the AP1 and AP2 clathrin adaptor complexes (*Schledzewski et al., 1999*). X-ray crystallography of purified coatomer components (*Watson et al., 2004*; *Yu et al., 2009*; *Lee and Goldberg, 2010*; *Hsia and Hoelz, 2010*; *Yu et al., 2012*; *Jackson et al., 2012*; *Ma and Goldberg, 2013*), as well as cryo-electron tomography (cryo-ET) of COPI-coated vesicles assembled *in vitro*,

have led to a complete molecular model of the COPI coat (*Faini et al., 2012*; *Dodonova et al., 2015*, *Dodonova et al., 2017*). Within the assembled coat, three copies of coatomer and six copies of Arf1 form a three-fold triad structure in which the coat-like and adaptor-like subcomplexes of coatomer form intertwined arches. Triads link together on the membrane surface to produce vesicles of variable size and shape.

Disassembly of the coat after budding is triggered by hydrolysis of Arf1-GTP, which is regulated by GTPase activating proteins (ArfGAPs) (*Popoff et al., 2011*). The dynamics of GTP hydrolysis and coat disassembly are poorly understood; it remains unclear whether Arf1 dissociates from the membrane before or together with coatomer, as well as whether the coat disassembles gradually or *en masse*.

The diversity of COPI function has led to the hypothesis that there are different types of COPI vesicles. Biochemical and microscopic studies have categorized COPI vesicles based on their donor compartment, cargo load, possible transport direction, isoform composition, and morphology (*Orci et al., 1997*; *Lanoix et al., 2001*; *Malsam et al., 2005*; *Moelleken et al., 2007*; *Donohoe et al., 2007*, *Donohoe et al., 2013*; *Bouchet-Marquis et al., 2008*). Based on conventional electron microscopy of the *Scherfellia dubia* Golgi, vesicles were grouped into COPIa, which have lightly stained luminal content and surround the *cis* region of the stack, and COPIb, which have strongly stained luminal content and are found near the *medial* Golgi. Vesicles budded from the final *trans* cisterna were characterized as a separate type of secretory vesicle based on their morphology (*Donohoe et al., 2007*, *Donohoe et al., 2013*). Cryo-electron microscopy of mammalian cell vitreous sections suggested that one group of COPI vesicles and buds has a 'spiky' coat morphology while a different group has a smoother, more homogeneous coat (*Bouchet-Marquis et al., 2008*). The molecular basis for the apparent heterogeneity of COPI vesicles is unknown.

We previously used cryo-electron tomography and subtomogram averaging (*Wan and Briggs, 2016*) to build a molecular model of the COPI coat budded *in vitro* from giant unilamellar vesicles using purified coat protein components (*Faini et al., 2012*; *Dodonova et al., 2015*, *Dodonova et al., 2017*). To investigate the structure, distribution and diversity of COPI within the cell, we combined cryo-electron tomography and subtomogram averaging with cryo-focused ion beam (cryo-FIB) milling of vitrified cells (*Schaffer et al., 2017*; *Marko et al., 2007*; *Rigort et al., 2012*). We imaged Golgi stacks within the native cellular environment of the genetically-tractable model organism, *Chlamydomonas reinhardtii* (*Jinkerson and Jonikas, 2015*). This unicellular green alga provides reproducible Golgi architecture (*Farquhar and Palade, 1981*), enabling comparative analysis across multiple cells, as well as superb cryo-EM imaging conditions that enhance the fidelity of structure determination *in situ*, within the cell (*Pfeffer et al., 2017*; *Freeman Rosenzweig et al., 2017*). Previously, *in situ* cryo-ET studies of this organism revealed the presence of ordered intracisternal arrays that may help maintain the architecture of the *C. reinhardtii* Golgi (*Engel et al., 2015*).

## Results and discussion

We collected a dataset of 29 cryo-electron tomograms, each containing a clearly-visible Golgi stack. Most Golgi stacks in our dataset consisted of nine cisternae that exhibited reproducible morphology (*Figure 1A–B*, *Video 1*). Luminal density progressively darkened through the *cis* and *medial* Golgi, while the cisterna centers narrowed and edges swelled. The final *trans* cisterna and the adjacent ballooning compartments of the trans-Golgi-network (TGN) both contained a translucent lumen (*Figure 1A*).

All three archetypal protein coats could be visually identified in the tomograms without any ambiguity: clathrin-coated vesicles were found in the vicinity of the TGN and were distinguished by their characteristic triskelion-based cage (*Figure 1C,C'*), while ER exit sites with COPII buds and vesicles were distinguished by their two-layered coat (*Figure 1D,D'*). Multiple COPI-coated vesicles and buds were found around the periphery of the Golgi cisternae and were discriminated from clathrin and COPII-coated membranes by the presence of a dense uniform coat (*Figure 1E–H*).

We extracted 267 buds and vesicles showing extensive COPI coats. We applied a reference-free subtomogram averaging workflow, as previously described (*Faini et al., 2012*), to determine the structure of the COPI coat *ab initio*, without making use of structures previously determined *in vitro*.

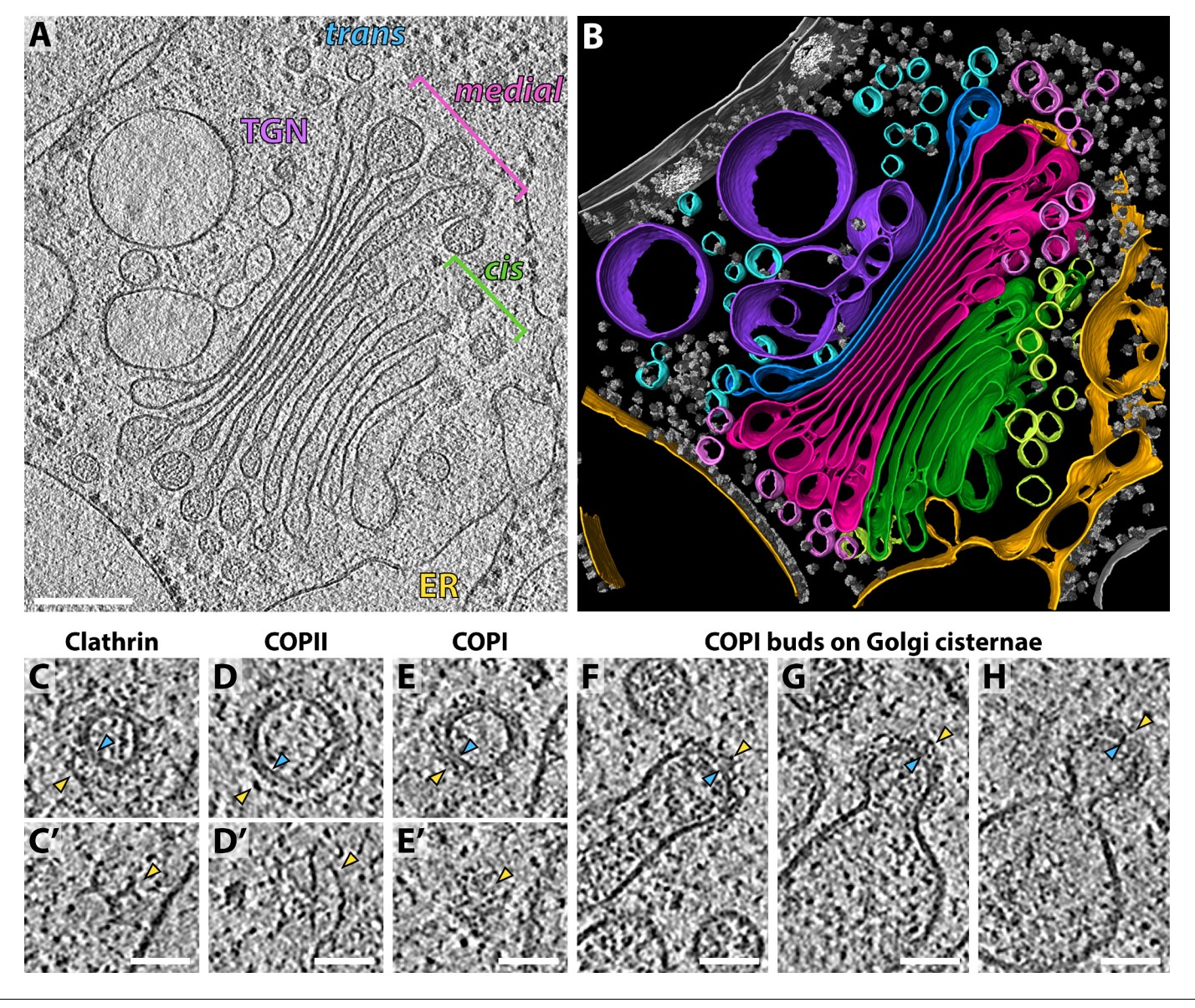

**Figure 1.** Molecular architecture of the *Chlamydomonas* Golgi apparatus and transport vesicles revealed by *in situ* cryo-ET. (**A**) A slice through a cellular tomogram containing a representative Golgi stack and (**B**) corresponding 3D segmentation, showing the native morphology of the ER (yellow), four *cis* cisternae (green), *cis* vesicles (light green), four *medial* cisternae (magenta), *medial* vesicles (light pink), the *trans* cisterna (blue), *trans* vesicles (light blue) and the TGN (purple). Other membranes, the nuclear envelope, nuclear pore complexes and ribosomes are shown in grey. (**C–E**) Slices through (**C, C'**) clathrin, (**D, D'**) COPII and (**E, E'**) COPI coated transport vesicles found within the tomograms. Panels C, D and E are slices through the centers of the vesicles. Panels C', D' and E' are top slices through the coats of the same vesicles, showing characteristic structures of (**C'**) clathrin triskelions, (**D'**) the triangular Sec13/31 COPII lattice, and (**E'**) the dense COPI coat. (**F–H**) Slices through three COPI buds in different stages of maturation. Yellow arrowheads: cytoplasmic boundary of the coat, blue arrowheads: vesicle or bud membrane. Scale bars: 200 nm in A-B, 50 nm in C-H.
DOI: https://doi.org/10.7554/eLife.32493.002

We divided the buds and vesicles into two half-datasets, which were processed independently (*Figure 2—figure supplement 1A*). The two resulting structures were compared by Fourier shell correlation and averaged together to generate a final structure from 3579 subtomograms (10,737 asymmetric units) with a resolution of 20 Å (*Figure 2—figure supplement 1B*). The features of the map are consistent with the measured resolution (*Figure 2A*).

The native *in situ* structure of COPI from *C. reinhardtii* is strikingly similar to structures previously determined from *in vitro* reconstituted COPI budding reactions using mouse proteins (*Figure 2—*

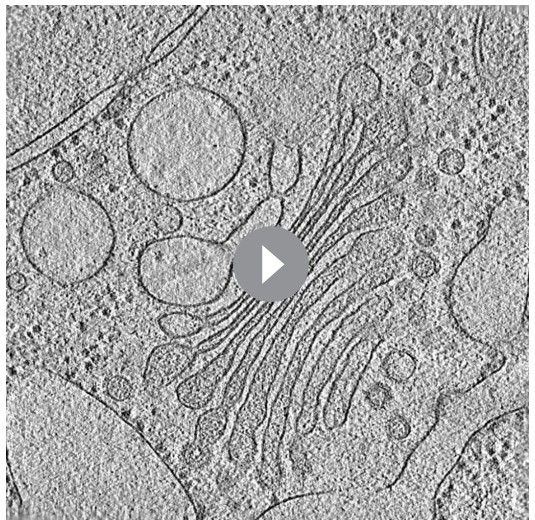

**Video 1.** *In situ* cryo-electron tomogram of the native *Chlamydomonas* Golgi. The movie slices back and forth in Z through the tomographic volume, then reveals the 3D segmentation, colored as in *Figure 1B*. The ER is yellow, four *cis* cisternae are green, *cis* vesicles are light green, four *medial* cisternae are magenta, *medial* vesicles are light pink, the *trans* cisterna is blue, *trans* vesicles are light blue and the TGN is purple. Other membranes, the nuclear envelope, nuclear pore complexes and ribosomes are shown in grey.

DOI: https://doi.org/10.7554/eLife.32493.003

figure supplement 2) (*Faini et al., 2012*; *Dodonova et al., 2015*, *Dodonova et al., 2017*). The pseudoatomic model based on the *in vitro* structure fits into our *in situ* structure as a rigid body (*Figure 2A*, *Video 2*). This shows that the architecture of the COPI coat described *in vitro*, in which the adaptor-like and coat-like subcomplexes do not form two layers but rather form an intertwined trimeric assembly, is the architecture present within cells, and that this architecture is highly conserved between two distantly-related species.

A closer comparison of the *in vitro* and *in situ* structures reveals that the *in situ* map has an additional density positioned on the luminal side of the membrane below the N-terminal β–propeller of the β'–COP subunit (*Figure 2B,C*; *Figure 2—figure supplement 2D,E*). The position of this density, on the luminal side of the membrane directly below the dilysine cargo-binding site in β'–COP, and its absence in the *in vitro* system where there is no cargo, suggests that it corresponds to bound cargo or cargo receptor transported by vesicles. We found no additional density below the equivalent cargo-binding site in the homologous N-terminal β–propeller of α–COP, indicating that α-COP binds fewer, smaller, or more flexible cargos than β'–COP.

There are therefore differences in cargo binding between α-COP and β'–COP. While it was previously suggested that α-COP and β'–COP have distinct selectivity towards KKXX and KXKXX

ER retrieval motifs (*Eugster et al., 2004*; *Jackson et al., 2012*), more recent work suggests that cargo specificity may depend on the −2 position of the motif (*Ma and Goldberg, 2013*). The motif binding pockets in *C. reinhardtii* are conserved with respect to other eukaryotes, and *C. reinhardtii* contains retrieved proteins with both KKXX and KXKXX motifs. We see two possible interpretations of our observation. The first is that α-COP and β'–COP have specificities towards particular motifs, and that proteins containing these motifs differ in their abundance, size, or flexibility. The second is that the distinct positions of α-COP and β'–COP within the coat lead to differences in the amount of cargo bound: for example, this could be due to the more inclined tilt of the β'–COP β–propeller relative to the membrane, or differences in the timing, duration or strength of the interaction between the two β-propellers and the membrane during coat assembly. In both cases, the difference in cargo binding could either be specific to *C. reinhardtii* or a widespread property.

We visualized the coatomer organization by placing a triangle at the position and orientation of each COPI triad in the analyzed vesicles and buds (*Figure 2D*). Consistent with previous *in vitro* observations, the overall vesicle coat organization is pleomorphic and the arrangement of the triads does not conform to any type of global symmetry. Instead, both *in vitro* and *in situ*, triads contact each other in four defined patterns, called linkages (*Faini et al., 2012*; *Dodonova et al., 2015*). Within the cell, we determined the structures of the three most common linkages, which showed good overall agreement with the previously obtained *in vitro* structures (*Figure 2—figure supplement 3*). In linkage IV, interactions between triads involve ε-COP and the C-terminal domain of α–COP. Although ε-COP, like the μ-homology domain of δ-COP, is dispensable for COPI function (*Arakel et al., 2016*), it forms a rigid, defined interaction in the linkage, suggesting that it plays a role in coat assembly *in vivo*.

We observed vesicles with varying degrees of coat completeness, but no fully complete coats; as previously described *in vitro*, each vesicle contained a gap in the coat at the site of scission (the

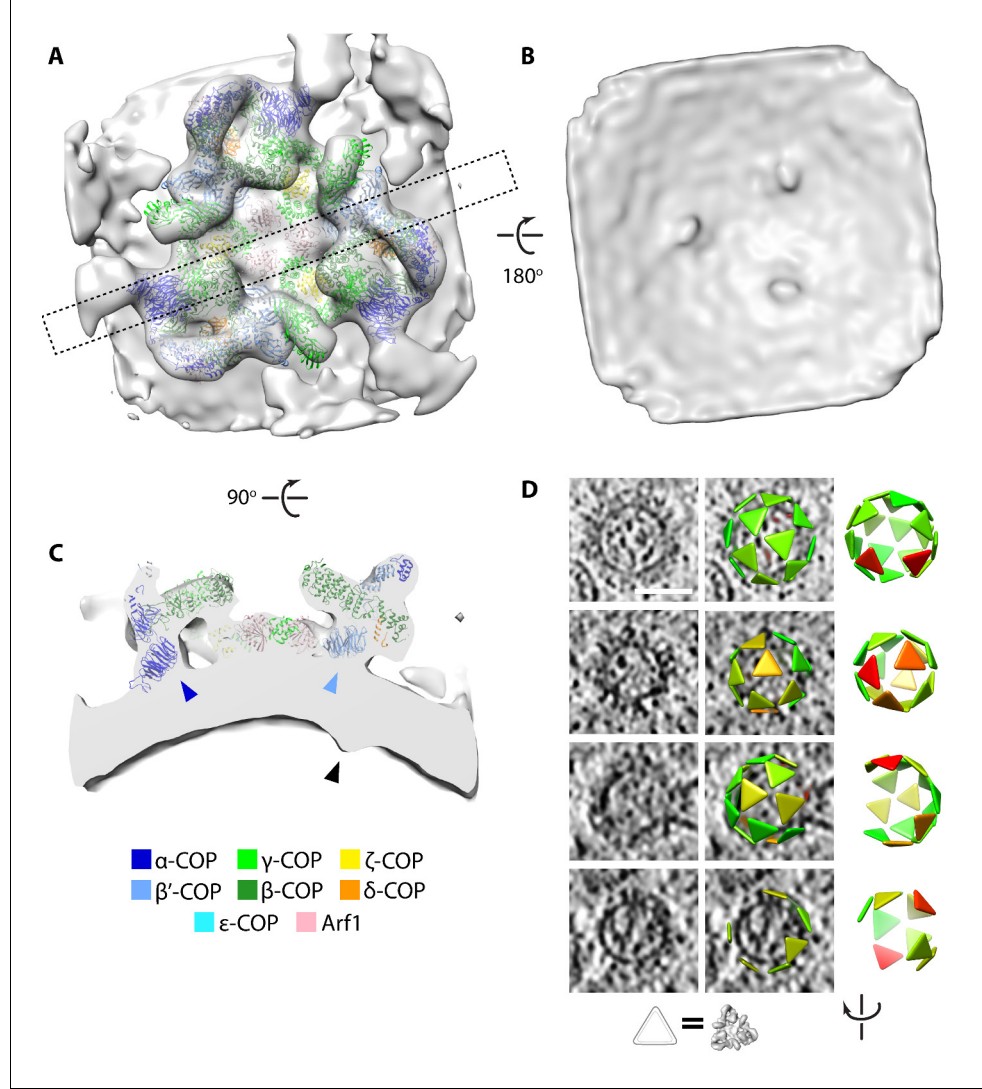

α-COP  γ-COP  ζ-COP
β'-COP  β-COP  δ-COP
ε-COP  Arf1

△ = 🝙

**Figure 2.** Determination of the native COPI coat structure and arrangement. (**A–C**) Isosurface view of the triad density map determined within the cell. (**A**) View from the cytoplasmic side, dashed rectangle indicates the position of the cross-section shown in panel C. (**B**) View from the vesicle lumen side, displaying three cargo density protrusions. (**C**) Cross-section view, showing the position of the cargo density (black arrowhead) relative to the positions of the N-terminal β–propellers of α–COP (dark blue arrowhead) and β'-COP (light blue arrowhead). (**D**) Examples of coat arrangement on vesicles and buds. In each row, left: tomographic slice; middle: tomographic slice overlaid with triangles at the positions and orientations of each triad (correspondence between triangle and structure is illustrated below), colored by similarity to the structure in panels A-C (green to red = high to low correlation); right: view of the triad arrangement rotated to orient the budding scar towards the viewer. The vesicle in the bottom row has an incomplete coat and is likely undergoing uncoating. Scale bar: 50 nm.
DOI: https://doi.org/10.7554/eLife.32493.004

The following figure supplements are available for figure 2:

**Figure supplement 1.** *In situ* COPI structure determination.
DOI: https://doi.org/10.7554/eLife.32493.005

**Figure supplement 2.** Comparison of *in situ* and *in vitro* structures.
DOI: https://doi.org/10.7554/eLife.32493.006

**Figure supplement 3.** Structures of the *in situ* triad linkages.
DOI: https://doi.org/10.7554/eLife.32493.007

'budding scar') (*Figure 2D*). A large population of uncoated vesicles was present in the ribosome exclusion zone immediately surrounding the Golgi. The luminal density and diameter distribution of these vesicles were the same as that of COPI-coated vesicles (*Figure 3—figure supplements 1* and *2*). Thus, the combination of location, size and density strongly suggests that the majority of them are uncoated COPI vesicles. We estimated coat completeness in XY slices through the centers of the vesicles in five selected Golgi tomograms (*Figure 3*, *Figure 3—figure supplement 1*). Since our data represent a random sampling of the COPI vesicle population in the proximity of the Golgi, the relative dynamics of vesicle uncoating can be inferred from the distribution of coat completeness. Coat disassembly begins after budding, and is fairly rapid but not catastrophic, since a significant number of vesicles at intermediate uncoating stages were observed. The percentage of uncoated vesicles indicates that uncoating is completed after approximately one third of the average vesicle lifetime (*Figure 3*). We could not determine whether individual coatomer complexes remain associated with the naked vesicles, or whether the uncoating process starts at the budding scar or at random coat positions.

Knowing the positions of individual COPI complexes in the cell enabled us to sort the complexes into discrete subpopulations based on their cellular location. To assess whether structural changes occur in the COPI coat during budding or initial uncoating, we calculated separate averages for buds and vesicles, for both triads and linkages. We did not detect any substantial differences in the structures (*Figure 3—figure supplement 3*). These observations suggest that the stoichiometry of coatomer subunits and Arf1 are the same during assembly and disassembly. This would not be the case if GTP hydrolysis led to loss of Arf1 from the coat either during or shortly after budding, as has been proposed (*Yang et al., 2002*; *Liu et al., 2005*), or if there were changes in the position or presence of any coatomer subunits during or after budding. Thus, we conclude that loss of Arf1 and coatomer disassembly occur simultaneously. The lack of any extra densities in the averages from buds and vesicles further indicates that additional factors such as ArfGAPs bind only transiently or at low stoichiometry.

COPI-coated vesicles changed their appearance during progression through the Golgi stack. We observed a gradual transition from lighter to darker vesicle luminal density through the stack that generally matched that of the donor cisternae, followed by an abrupt change to a translucent lumen in some vesicles found near the *trans* Golgi (*Figure 1A*, *Figure 4A,B*). Interestingly, COPI coated buds were also found on the translucent TGN compartments (*Figure 4—figure supplement 1*). We consider it likely that these buds mediate retrograde transport towards the *trans* Golgi, but cannot rule out that they yield anterograde secretory vesicles such as those previously suggested to exist in algae (*Donohoe et al., 2007*). While translucent buds were found on the TGN and some regions of the *trans* Golgi, most buds emerging from the translucent *trans* cisterna had darker luminal density that was similar to *medial* Golgi vesicles (*Figure 4—figure supplement 1*), indicating enrichment of cargo in the *trans* buds. The change in observed luminal density (which correlates directly with density observed in cryo-EM) through the Golgi stack indicates a progressive increase and then decrease in the amount of material being transported, consistent with the predominant vesicle cargo being resident Golgi proteins (*Donohoe et al., 2013*).

There was a gradual increase in the diameter of buds and vesicles from the *cis* through the

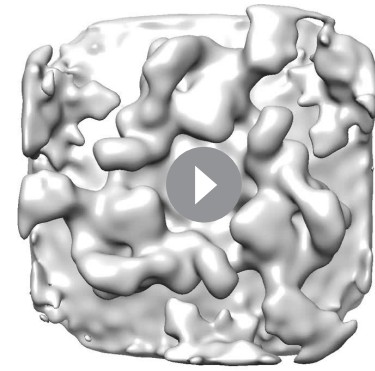

**Video 2.** Overview of the *in situ* COPI structure and bound cargo density. The movie shows an isosurface view of the native COPI triad structure. The PBD model 5A1U, derived from a structure of COPI determined *in vitro* (*Dodonova et al., 2015*), is fit into the structure as a rigid body. The isosurface is initially shown from the cytoplasmic side, before it is rotated 90° and sliced perpendicular to the membrane to show cargo density residing beneath the N-terminal β-propeller of β'-COPI but not beneath the analogous domain of α-COP. The isosurface is then rotated a further 90° and viewed from the vesicle lumen side to show three symmetric cargo densities.
DOI: https://doi.org/10.7554/eLife.32493.008

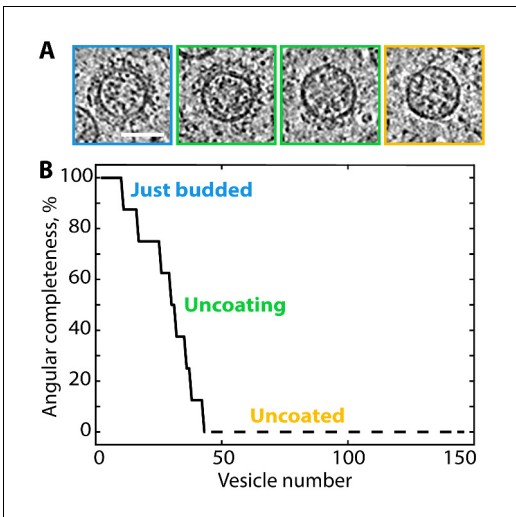

**Figure 3.** Ranked plot of COPI vesicle completeness indicates a rapid and processive uncoating process. (A) Central slices through COPI vesicles at different stages of uncoating (from left to right). (B) Vesicles were sorted according to the angular completeness of the coat, as measured from central slices (see Materials and methods). Scale bar: 50 nm.

DOI: https://doi.org/10.7554/eLife.32493.009

The following figure supplements are available for figure 3:

**Figure supplement 1.** Examples of coated and uncoated COPI vesicles from five Golgi stacks.
DOI: https://doi.org/10.7554/eLife.32493.010

**Figure supplement 2.** Diameter distributions of coated and uncoated COPI vesicles.
DOI: https://doi.org/10.7554/eLife.32493.011

**Figure supplement 3.** Comparison of the coat structures derived from buds and vesicles.
DOI: https://doi.org/10.7554/eLife.32493.012

*medial* and *trans* Golgi (*Figure 4—figure supplement 2*). The vesicles with a translucent lumen had a more variable diameter distribution. The increasing vesicle diameter may result from the swelling of the *medial* and *trans* cisterna edges, causing the curvature of the donor membrane to decrease. Similarly, the variable donor membrane curvature in the TGN likely yields a variety of translucent vesicle sizes.

We separated vesicles and buds into three groups ('*cis*', '*medial/trans*', and '*trans*/TGN'; *Figure 4A,B*) based on their position and luminal density (see Materials and methods). We averaged the subtomograms in each group to obtain structures of the COPI coat from three different Golgi regions (*Figure 4C–E*, *Figure 4—figure supplement 3*). The coat structure is highly similar between the three regions except for the luminal cargo/cargo-receptor density, which is less prominent in the average obtained from the translucent buds and vesicles of the *trans* Golgi and TGN. This suggests that vesicles in this region bind less, or less bulky, cargo in the vicinity of the β-propellers of β'-COP, which contain a dilysine cargo-binding site.

In the structures from each region, two peaks of bilayer density were resolved beneath the COPI coat (*Figure 4C*). Measurement of the distance between the peaks revealed a gradual increase in membrane thickness from *cis* to *trans* Golgi vesicles and buds (*Figure 4F,G*; *Figure 4—figure supplement 4*) (we note that absolute bilayer thickness cannot be measured from this data, see Materials and methods). We also measured a general increase in bilayer thickness in non-coated membranes of the *medial* and *trans* Golgi cisternae (*Figure 4H*, *Figure 4—figure supplements 5* and *6*). These observations are in good agreement with the hypothesis that changes in lipid composition create a gradient of membrane thickness that increases from the ER towards the plasma membrane, which has been suggested to play a role in transmembrane protein sorting along the secretory pathway (*Mitra et al., 2004*; *Sharpe et al., 2010*).

We observed that the morphology of COPI vesicles, including the luminal density, membrane thickness and vesicle size, change through the Golgi stack in a manner that reflects corresponding changes in the morphology of the donor cisterna. While vesicle morphology is inherited from the donor membrane, the composition of the vesicle differs from that of the donor cisterna. The specifically bound cargo under the β'–COP beta propellers, as well as the dark luminal density of buds on the translucent *trans* cisterna, both confirm that proteins are selectively incorporated. In contrast to the observed morphological changes through the stack, we found no indication of structurally distinct COPI vesicle subclasses; the structure of the sorting machinery is the same in all buds and vesicles.

This study shows that it is possible to determine the native structure of the COPI coat, without making use of previous structural information, directly within the cell, avoiding the artifacts associated with sample preparation for more traditional structural biology techniques. The tomograms represent snapshots of the dynamic cellular environment in its native state, and information from this

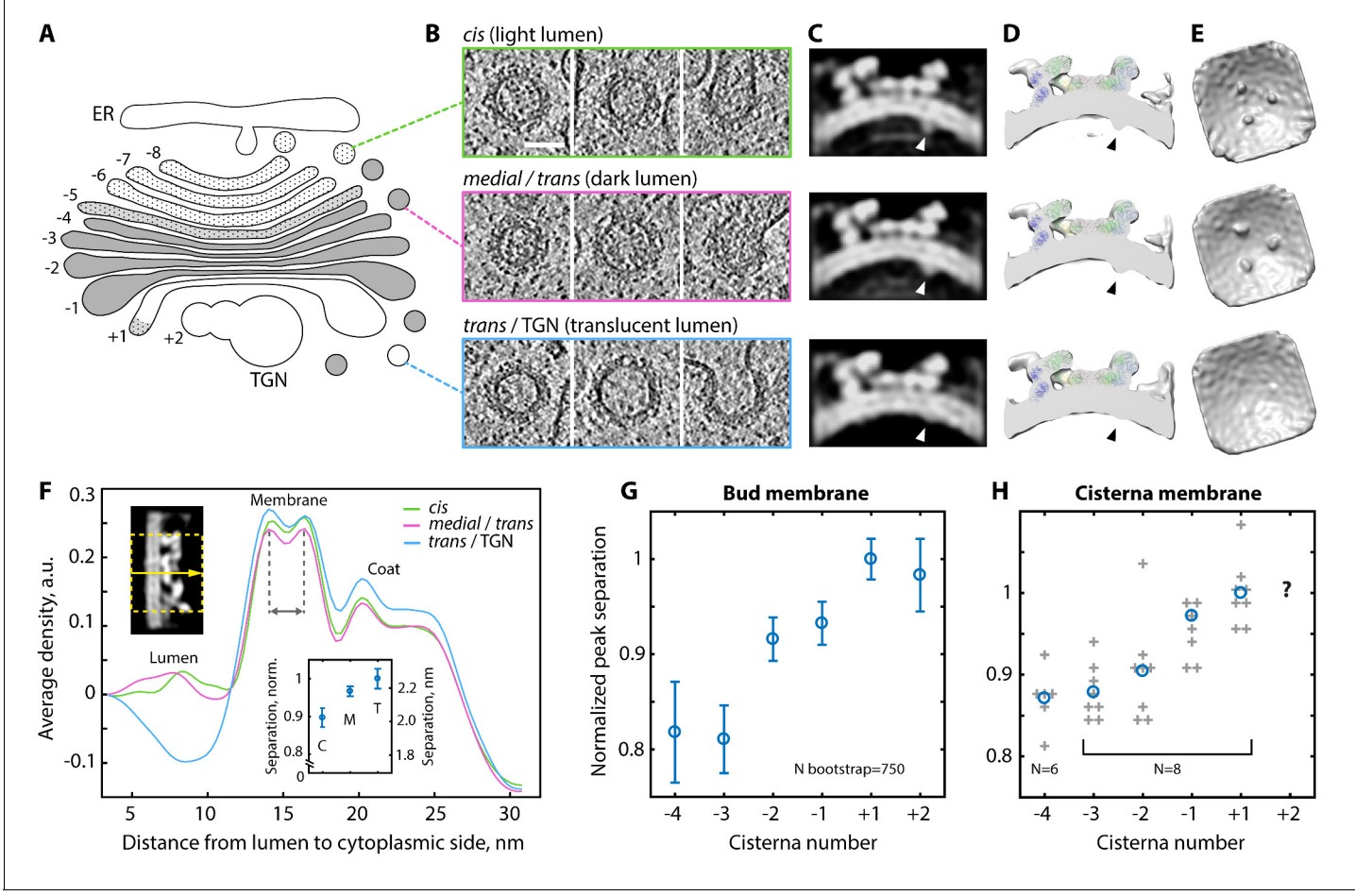

**Figure 4.** Variability of the COPI structure and membrane thickness throughout the Golgi. (A) Schematic of the *C. reinhardtii* Golgi showing cisternae with *cis* (−8 to −6), intermediate (−5), *medial* (−4 to −1), and *trans* morphology (+1) and the TGN (+2). Cisternae are numbered relative to the transition from dark to translucent lumen (between −1 and +1). (B) Examples of buds and vesicles from three groups defined by position and luminal density (see Materials and methods). Top row: '*cis*', middle row: '*medial/trans*', bottom row: '*trans*/TGN'. (C–E) ~24 Å structures of the coat determined from the three groups. (C) Orthographic slices and (D) cross-section isosurface views through the averages. Arrowheads: cargo. (E) Isosurface views from inside the vesicle. While the coatomer structure remains constant, the '*trans*/TGN' structure lacks bound cargo. (F) Density profiles through the three COPI structures. Top left inset: example orthographic slice through a straightened structure, dashed yellow rectangle and arrow: averaged region and direction of the profile. Bottom inset: bootstrapped mean bilayer leaflet separation measured for the '*cis*' {C}, '*medial/trans*' {M} and '*trans*/TGN' {T} averages, error bars: bootstrapped SD. (G–H) Membrane bilayer thickness in (G) COPI buds and (H) non-coated membrane regions from individual *medial* and *trans* Golgi cisternae. In G, mean and error bars (SD) were bootstrapped (see Materials and methods). In H, grey crosses: individual cisternae, blue circles: means, TGN measurements were not obtained. Scale bar: 50 nm.

DOI: https://doi.org/10.7554/eLife.32493.013

The following figure supplements are available for figure 4:

**Figure supplement 1.** COPI bud and vesicle morphology varies across the Golgi stack.
DOI: https://doi.org/10.7554/eLife.32493.014

**Figure supplement 2.** COPI vesicle and late bud diameter variation across the Golgi stack.
DOI: https://doi.org/10.7554/eLife.32493.015

**Figure supplement 3.** Structure determination of COPI obtained from different Golgi regions.
DOI: https://doi.org/10.7554/eLife.32493.016

**Figure supplement 4.** Density profiles through the averages derived from buds in individual *trans* and *medial* Golgi cisternae.
DOI: https://doi.org/10.7554/eLife.32493.017

**Figure supplement 5.** Membrane bilayer thickness measurements in individual cisternae of *medial* and *trans* Golgi.
DOI: https://doi.org/10.7554/eLife.32493.018

**Figure supplement 6.** Membrane bilayer profiles measured in individual cisternae of *medial* and *trans* Golgi.
DOI: https://doi.org/10.7554/eLife.32493.019

cellular context gives additional insights into COPI function. We believe that the ability to determine structure directly within the cellular environment can transform cell and structural biology.

# Materials and methods

## Key resources table

| Reagent type (species) or resource | Designation | Source or reference | Identifiers | Additional information |
|---|---|---|---|---|
| strain, strain background (*Chlamydomonas reinhardtii*) | mat3-4 mt+ | PMID: 11445540, DOI: 10.1101/gad.892101 | CC-3994, Chlamydomonas Resource Center | small cells improve vitrification |
| software, algorithm | SerialEM | PMID: 16182563, DOI: 10.1016/j.jsb.2005.07.007 | | cryo-ET tilt-series acquisition |
| software, algorithm | MotionCor2 | PMID: 28250466, DOI: 10.1038/nmeth.4193 | | frame alignment |
| software, algorithm | IMOD | PMID: 8742726, DOI: 10.1006/jsbi.1996.0013 | | tomogram alignment, reconstruction |
| software, algorithm | Amira | FEI/Thermo Fisher Sceintific | | 3D segmentation of tomogram volumes |
| software, algorithm | Matlab | Mathworks | | subtomogram averaging, data analysis |
| software, algorithm | TOM package for Matlab | PMID: 15721576, DOI: 10.1016/j.jsb.2004.10.006 | | subtomogram averaging |
| software, algorithm | Av3 package for Matlab | PMID: 15774580, DOI: 10.1073/pnas.0409178102 | | subtomogram averaging |
| software, algorithm | RELION with helical processing | PMID: 28193500, DOI: 10.1101/095034 | | averaging of 2D membrane profiles |
| software, algorithm | USCF Chimera | PMID: 15264254, DOI: 10.1002/jcc.20084 | | visualization |

## Cell culture and sample preparation

Cell culture and sample preparation were performed as previously described (*Engel et al., 2015*). *Chlamydomonas reinhardtii* strain *mat3-4* (*Umen and Goodenough, 2001*), characterized by a smaller cell size that aids vitrification, was acquired from the Chlamydomonas Resource Center (University of Minnesota) and grown with constant light (~90 µmol photons $m^{-2}$ $s^{-1}$) and normal atmosphere aeration in Tris-acetate-phosphate (TAP) medium. Plunge-freezing and cryo-focused ion beam milling were performed as previously described (*Schaffer et al., 2017*). Cells were blotted onto carbon-coated 200-mesh copper grids (Quantifoil Micro Tools), which were plunged into a liquid ethane-propane mixture using a Vitrobot Mark 4 (FEI). Frozen grids were mounted into modified Autogrids (FEI) and loaded into either a Scios (FEI) or a Quanta 3D FEG (FEI) FIB/SEM microscope. Samples were coated with thin layer of organometallic platinum using the *in situ* gas injection system (GIS, FEI). Thin lamellae containing the cellular interior were milled using the $Ga^+$ beam at 30 kV and a shallow angle of 8°−12°.

## Cryo-electron tomography

Tomograms were collected with SerialEM software (*Mastronarde, 2005*) on a 300 kV Titan Krios (FEI) equipped with a postcolumn energy filter (968 Quantum, Gatan) and a K2 Summit direct electron detector (Gatan) operated in movie mode at 12 frames per second. Tilt series were acquired over an angular range of approximately −60° to +60°, using a bi-directional tilt scheme with a 2° increment. Cumulative electron dose was kept under ~100 $e^-/Å^2$, the object pixel size was 3.42 Å, and defocus ranged from −4 to −6 um. Each tomogram was acquired from a separate cell, and thus each is a biological replicate. Around five different cell cultures were used in total. During 16 microscopy sessions, a total of 60 tomograms were acquired that contained Golgi or coated vesicle structures. These tomograms were then sorted by image quality and tilt-series alignment precision, yielding 29 tomograms that were used for analysis, including nine described in a previously published study of Golgi intracisternal arrays (*Engel et al., 2015*).

## Subtomogram averaging

K2 image frames were aligned with MotionCor2 (*Zheng et al., 2017*) to correct for beam-induced motion. Cumulative electron dose-dependent low-pass filters were calculated as previously determined, assuming a uniformly distributed total dose of 60 e⁻/Å² (*Grant and Grigorieff, 2015*), and were applied to the tilt series before processing. Contrast transfer-function (CTF) correction of individual tilt images was performed using *ctfphaseflip* implemented in IMOD (*Xiong et al., 2009*). Tilt series were aligned in IMOD using patch-tracking and reconstructed using weighted backprojection (*Kremer et al., 1996*).

Tomograms were visualized in Amira software (FEI). We localized 267 COPI vesicles and buds, measured their membrane-to-membrane diameters and determined their central coordinates. Within cells, the mean (±SD) diameter of late buds and vesicles was 56 ± 6 nm (*Figure 3—figure supplement 2*), approximately 10 nm wider than that of vesicles budded *in vitro*. Subtomogram alignment and averaging was performed using Matlab (Mathworks) scripts derived from the TOM (*Nickell et al., 2005*) and Av3 (*Förster et al., 2005*) toolboxes, essentially as described (*Dodonova et al., 2015*). The diameters and coordinates of buds and vesicles were used to define the starting positions and orientations of subtomograms. The positions were placed as evenly distributed points (spacing 10 nm) on the surface of a sphere defined by the vesicle center and diameter. Initial psi and theta angles, which define out-of-plane rotation, were assigned as the normal to the sphere surface. Phi angles defining in-plane rotation were assigned randomly.

To generate the initial reference, we selected 64 vesicles and late buds that were completely embedded within the lamellae, had diameters that were close to average and had extensive coats as judged by visual inspection. This set was split into two independent halves, which were processed separately. Subtomograms were extracted from twice-binned volumes with a pixel size of 13.68 Å. Initial smooth reference averages for the first round of alignment were generated based on the assigned Euler angles, and subtomograms were aligned iteratively using the result of the previous round as a reference. After convergence, both independent averages contained an identical three-fold symmetrical feature in different positions. This feature was brought to the center of both references and additional alignments were performed with three-fold symmetry applied (*Figure 2—figure supplement 2A*). After centering the structures and applying threefold symmetry, the two references looked the same and were similar to the structure of the COPI coat obtained *in vitro* (*Figure 2—figure supplement 2A*). The two structures were then rotated to the same orientation and used as starting references for alignment of the full dataset.

The remaining vesicles and buds were divided between the two half-datasets. Subtomograms were extracted as described above and aligned to the starting references. After alignment, we removed subtomograms with low cross-correlation coefficients to the reference. Overlapping subtomograms were removed based on the distance threshold of 17.7 nm (13 pixels). The final datasets had 1713 and 1866 subtomograms (5139 and 5598 asymmetric units). The structures were further refined by processing subtomograms extracted from once-binned (pixel size 6.84 Å) and unbinned (pixel size 3.42 Å) volumes. Prior to the resolution measurement, each reference was masked with a soft-edged cylindrical mask that excluded the membrane. Resolution was measured using mask-corrected phase-randomized Fourier shell correlation (FSC) (*Figure 2—figure supplement 1B*) (*Chen et al., 2013*). The final density map was obtained by averaging the structures from the two half-datasets together. The final structure was low-pass filtered to the determined resolution, reweighted by dividing by the sum of its CTFs, and sharpened with a B-factor of −2000 (*Rosenthal and Henderson, 2003*). Map visualization and fitting of PDB structures were performed using Chimera software (*Pettersen et al., 2004*). The *in vitro* pseudoatomic model 5A1U was fit into our *in situ* structure as rigid body.

## Lattice map and linkage structure analysis

Triad positions and orientations were visualized using a custom chimera plugin. No completely coated vesicles were found. Some of the vesicles were not completely contained within the FIB lamella and, in other cases, some triads may not have been successfully identified during the alignment procedure. For the remaining COPI-coated cellular vesicles, we could determine that the coat is incomplete, either because areas of uncoated membrane are clearly visible in the tomogram, or because the arrangement of triads would not allow the vesicle to be closed by the addition of more

triads (*Figure 2D*). The absence of any complete vesicle coats could reflect rapid partial uncoating after budding, or result from a budding scar, where the coat is never completed at the bud neck. The absence of complete vesicles in *in vitro* reconstitution reactions (*Faini et al., 2012*), which are performed in the presence of non-hydrolysable GTP analogues, suggests that the gap in the coat represents a budding scar.

Linkage position identification and structure determination were performed as previously described (*Faini et al., 2012*). Briefly, triad positions were visualized and example sets of triads representing each linkage were manually picked. These sets were used to calculate the relative rotations and translations between the triads in each linkage and search the dataset for positions matching these patterns. This automated search identified 946 occurrences of linkage I, 1013 occurrences of linkage II, and 271 occurrences of linkage IV. Linkage III has an inherent high curvature and was observed *in vitro* to be present mainly on small vesicles (*Faini et al., 2012*). Because the average diameter of cellular vesicles is larger than that of *in vitro* vesicles, this arrangement of triads was only found on a few small vesicles within the cell (*Figure 2—figure supplement 3*).

To determine the structures of linkages I, II and IV, subtomograms were extracted from once-binned volumes (pixel size 6.84 Å) at the identified positions and divided into two half-sets. Euler angles were assigned to each pattern. We randomized the third Euler angle and ran an in-plane iterative rotational alignment on the first half-dataset, regenerating the linkage structure. This reference was low-pass filtered to 55 Å and used as a starting reference to align the second half-dataset.

Resolution was measured as described above and was determined to be 32–33 Å for all linkages. PDB pseudoatomic models 5NZT, 5NZU, and 5NZV from the *in vitro* study were used for fitting. We previously found that linkage I appears approximately three-fold symmetric at low resolution (*Faini et al., 2012*), but is asymmetric at higher resolution (*Dodonova et al., 2015*). For the *in situ* linkage I average, we were unable to resolve the asymmetry, and the structure was similar to that previously determined at low resolution. The structure of linkage II was also similar to that observed at low resolution. The density from the μ-homology domains of δ-COP located at the center of the pattern were not well resolved, suggesting that these domains are flexible in their position.

## Measurement of vesicle coat completeness

To measure the distribution of vesicle coat completeness, we identified 42 coated vesicles and 103 uncoated vesicles in five selected Golgi stacks (*Figure 3—figure supplement 1*). 13 tomographic slices around the vesicle center were averaged, and the resulting image was overlaid with a mask dividing it into eight sectors. The number of sectors containing coated membrane was recorded for each vesicle and used to calculate the angular completeness (*Figure 3*). While there are no complete vesicles in the dataset, mostly-complete vesicles may appear to be complete in a central slice. Thus, the measurements can systematically overestimate the completeness of mostly-complete vesicles, and underestimate the completeness of mostly-incomplete vesicles. The ratio of vesicles with coat to uncoated vesicles was ~1:2. We estimated the lifetime of coated vesicles as 1/3 of the total vesicle lifetime.

## Morphological annotation and structure variation analysis

The dataset of 267 vesicles and buds was visually annotated. Each vesicle or bud was assessed for budding stage (early bud, late bud with neck, vesicle), luminal density and position within the Golgi stack. Cisternal morphology was defined according to the scheme shown in *Figure 4A*. Only vesicles and buds whose positions could be confidently defined were used for generating separate averages from the *cis*, *medial*/*trans* and *trans*/TGN regions.

For structural analysis across the Golgi stack, we aimed to separate vesicles and buds into three groups that had maximal morphological and positional variation. For the '*cis*' average, we included all buds on the cisternae with *cis* morphology and position (usually the first four cisternae from *cis* side, which had uniform thickness and light luminal density) and all vesicles residing near the first two cisternae of the Golgi stack and between the Golgi and ER. Some of these vesicles were found in close proximity to the ER membrane. For the '*medial*/*trans*' average, we took all buds from cisternae with *medial* morphology (narrow region at the cisterna center, swollen periphery, dense lumen) and all vesicles not assigned to the '*cis*' and '*trans*/TGN' classes. Most of the buds on the last *trans* cisterna were included in the '*medial*/*trans*' average due to their dark luminal density (*Figure 4—*

*figure supplement 1*). For the '*trans/TGN*' average, we took vesicles and buds with a characteristic translucent lumen. These selected buds mostly emanated from TGN compartments. '*Trans/TGN*' vesicles resided near the *trans* Golgi and TGN, and were easily distinguished from '*medial/trans*' vesicles by lumen density. There was no difference in the distribution of cross-correlation values between each group of subtomograms and the average structure (*Figure 4—figure supplement 3*), suggesting that there are no large changes in the coat organization through the Golgi.

To measure membrane thickness in buds from individual cisternae, we defined the cisterna number relative to the first compartment with a translucent lumen, as diagrammed in *Figure 4A*. This compartment, which normally corresponded to the *trans* cisterna but occasionally the TGN (*Figure 4—figure supplement 5*), was numbered +1. Preceding cisterna were labeled according to their position as −1, −2, −3, −4, −5. Translucent compartments that followed +1, corresponding to swollen and heterogeneously shaped TGN, were labeled +2.

## Measurement of membrane thickness in vesicles and buds

Comparison of orthographic slices through the *cis, medial/trans* and *trans*/TGN structures suggested that there are differences in membrane bilayer thickness and curvature through the Golgi. We analyzed the membrane thickness change in more detail by generating a density profile through 'straightened' references. The membrane thickness was measured from straightened subtomogram averages of the vesicle/bud groups defined above: *cis, medial/trans* and *trans*/TGN (*Figure 4F*), as well as averages of buds from individual cisternae of the *medial* and *trans* Golgi (*Figure 4G*, *Figure 4—figure supplement 4*). The averages were low-pass filtered to 25 Å and sharpened with a B-factor of −2000. Volumes were straightened using radial orthographic projection as previously described (*de Marco et al., 2010*), and the average density profile was determined along the perpendicular vector to the straightened reference's membrane (*Figure 4F*, top left inset). The distance between the two peaks of the membrane bilayer was measured. For each subset, we then applied a bootstrap approach in which we repeated the procedure described above (from averaging to measurement) 750 times on sets of subtomograms equal in size to the original subset but randomly sampled from it with replacements. From these 750 profiles, we determined the mean and standard deviation of the membrane thickness. Data analysis was performed in Matlab (Mathworks).

## Measurement of membrane thickness in Golgi cisternae

To measure the membrane thickness in stacked, non-coated Golgi cisternae, we selected eight Golgi stacks in which the bilayers could be clearly observed and the cisternae were perpendicular to the imaging plane throughout the tomographic volume (*Figure 4—figure supplement 5A*). We extracted XY slices from the tomogram, then extracted 20 × 20 boxes along a bilayer, which were aligned and averaged using elements of the RELION helical reconstruction workflow (*He and Scheres, 2017*) to obtain a well-resolved 2D bilayer average (*Figure 4—figure supplement 5B*) for each cisterna. The number of particles for each average ranged from 100 to 300. The membrane density profile along the perpendicular vector to the bilayer was extracted using Fiji (*Schindelin et al., 2012*) and the peak-peak distance was measured (*Figure 4—figure supplement 6*).

The membrane profile is modulated by a number of effects that are difficult to assess including the contrast transfer function of the microscope, the charge gradient across the bilayer, the presence or absence of associated proteins and the exact orientation of bilayer relative to the beam (in case of 2D average). For this reason, while measurements can be compared within one experiment, it is important to note that the peak-peak distance does not correspond to the absolute bilayer thickness and cannot be compared to measurements performed in different ways such as those from the subtomogram averages.

## Acknowledgements

EM maps are deposited in the Electron Microscopy Data Bank (accession codes EMD-3968, EMD-3969, EMD-3970, EMD-3971, EMD-3972, EMD-3973, EMD-3974, EMD-3975, EMD-3976, EMD-3977). We thank K Qu for providing the Chimera plugin, S Munro and F Wieland for comments on the manuscript and W Wan for discussions. Our work was technically supported by EMBL IT services. JAGB and colleagues were funded by the European Molecular Biology Laboratory, the Chica and

Heinz Schaller Stiftung, and grants from the Deutsche Forschungsgemeinschaft within SFB1129 (Z2). WB and colleagues were funded by the Max Planck Society and the Deutsche Forschungsgemeinschaft Excellence Cluster Center for Integrated Protein Science Munich (CIPSM) and SFB-1035/Project A01.

## Additional information

### Funding

| Funder | Grant reference number | Author |
|---|---|---|
| Deutsche Forschungsgemeinschaft | SFB-1035/Project A01 | Wolfgang Baumeister |
| Deutsche Forschungsgemeinschaft | SFB1129 (Z2) | John AG Briggs |
| Medical Research Council | MC_UP_1201/16 | John AG Briggs |

The funders had no role in study design, data collection and interpretation, or the decision to submit the work for publication.

### Author contributions

Yury S Bykov, Conceptualization, Formal analysis, Investigation, Visualization, Methodology, Writing—original draft; Miroslava Schaffer, Svetlana O Dodonova, Investigation, Methodology, Writing—review and editing; Sahradha Albert, Investigation, Writing—review and editing; Jürgen M Plitzko, Supervision, Funding acquisition, Project administration, Writing—review and editing; Wolfgang Baumeister, Conceptualization, Supervision, Funding acquisition, Project administration, Writing—review and editing; Benjamin D Engel, Conceptualization, Supervision, Investigation, Visualization, Methodology, Writing—original draft, Project administration; John AG Briggs, Conceptualization, Supervision, Funding acquisition, Methodology, Writing—original draft, Project administration

### Author ORCIDs

Yury S Bykov https://orcid.org/0000-0003-2959-4108
Svetlana O Dodonova https://orcid.org/0000-0002-5002-8138
Jürgen M Plitzko https://orcid.org/0000-0002-6402-8315
Benjamin D Engel https://orcid.org/0000-0002-0941-4387
John AG Briggs http://orcid.org/0000-0003-3990-6910

### Decision letter and Author response

Decision letter https://doi.org/10.7554/eLife.32493.065
Author response https://doi.org/10.7554/eLife.32493.066

## Additional files

### Supplementary files
• Transparent reporting form
DOI: https://doi.org/10.7554/eLife.32493.020

### Major datasets
The following datasets were generated:

| Author(s) | Year | Dataset title | Dataset URL | Database, license, and accessibility information |
|---|---|---|---|---|
| Yury S Bykov, Miroslava Schaffer, Svetlana O Dodonova, Sahradha Albert, Jürgen M Plitzko, Wolfgang Baumeister, Benjamin D Engel, John AG Briggs | 2017 | The in situ structure of the Chlamydomonas COPI coat: average from vesicles | http://www.ebi.ac.uk/pdbe/entry/emdb/EMD-3970 | Publicly available at the EBI European Nucleotide Archive (accession no: EMD-3970) |
| Yury S Bykov, Miroslava Schaffer, Svetlana O Dodonova, Sahradha Albert, Jürgen M Plitzko, Wolfgang Baumeister, Benjamin D Engel, John AG Briggs | 2017 | The in situ structure of the Chlamydomonas COPI coat: average from the cis-Golgi region | http://www.ebi.ac.uk/pdbe/entry/emdb/EMD-3971 | Publicly available at the EBI European Nucleotide Archive (accession no: EMD-3971) |
| Yury S Bykov, Miroslava Schaffer, Svetlana O Dodonova, Sahradha Albert, Jürgen M Plitzko, Wolfgang Baumeister, Benjamin D Engel, John AG Briggs | 2017 | The in situ structure of the Chlamydomonas COPI coat: average from the medial/trans-Golgi region | http://www.ebi.ac.uk/pdbe/entry/emdb/EMD-3972 | Publicly available at the EBI European Nucleotide Archive (accession no: EMD-3972) |
| Yury S Bykov, Miroslava Schaffer, Svetlana O Dodonova, Sahradha Albert, Jürgen M Plitzko, Wolfgang Baumeister, Benjamin D Engel, John AG Briggs | 2017 | The in situ structure of the Chlamydomonas COPI coat: average from the trans-Golgi/TGN region | http://www.ebi.ac.uk/pdbe/entry/emdb/EMD-3973 | Publicly available at the EBI European Nucleotide Archive (accession no: EMD-3973) |
| Yury S Bykov, Miroslava Schaffer, Svetlana O Dodonova, Sahradha Albert, Jürgen M Plitzko, Wolfgang Baumeister, Benjamin D Engel, John AG Briggs | 2017 | The in situ structure of the Chlamydomonas COPI coat: average of all data | http://www.ebi.ac.uk/pdbe/entry/emdb/EMD-3968 | Publicly available at the EBI European Nucleotide Archive (accession no: EMD-3968) |
| Yury S Bykov, Miroslava Schaffer, Svetlana O Dodonova, Sahradha Albert, Jürgen M Plitzko, Wolfgang Baumeister, Benjamin D Engel, John AG Briggs | 2017 | The in situ structure of the Chlamydomonas COPI coat: average from buds | http://www.ebi.ac.uk/pdbe/entry/emdb/EMD-3969 | Publicly available at the EBI European Nucleotide Archive (accession no: EMD-3969) |
| Yury S Bykov, Miroslava Schaffer, Svetlana O Dodonova, Sahradha Albert, Jürgen M Plitzko, Wolfgang Baumeister, Benjamin D Engel, John AG Briggs | 2017 | The in situ structure of the Chlamydomonas COPI coat linkage I | http://www.ebi.ac.uk/pdbe/entry/emdb/EMD-3974 | Publicly available at the EBI European Nucleotide Archive (accession no: EMD-3974) |
| Yury S Bykov, Miroslava Schaffer, Svetlana O Dodonova, Sahradha Albert, Jürgen M Plitzko, Wolfgang Baumeister, Benjamin D Engel, John AG Briggs | 2017 | The in situ structure of the Chlamydomonas COPI coat linkage II | http://www.ebi.ac.uk/pdbe/entry/emdb/EMD-3975 | Publicly available at the EBI European Nucleotide Archive (accession no: EMD-3975) |

| Yury S Bykov, Miroslava Schaffer, Svetlana O Dodonova, Sahradha Albert, Jürgen M Plitzko, Wolfgang Baumeister, Benjamin D Engel, John AG Briggs | 2017 | The in situ structure of the Chlamydomonas COPI coat linkage IV | http://www.ebi.ac.uk/pdbe/entry/emdb/EMD-3976 | Publicly available at the EBI European Nucleotide Archive (accession no: EMD-3976) |
| Yury S Bykov, Miroslava Schaffer, Svetlana O Dodonova, Sahradha Albert, Jürgen M Plitzko, Wolfgang Baumeister, Benjamin D Engel, John AG Briggs | 2017 | Cryo-electron tomogram of the Chlamydomonas reinhardtii Golgi apparatus | http://www.ebi.ac.uk/pdbe/entry/emdb/EMD-3977 | Publicly available at the EBI European Nucleotide Archive (accession no: EMD-3977) |

The following previously published datasets were used:

| Author(s) | Year | Dataset title | Dataset URL | Database, license, and accessibility information |
| --- | --- | --- | --- | --- |
| Dodonova SO, Aderhold P, Kopp J, Ganeva I, Röhling S, Hagen WJH, Sinning I, Wieland F, Briggs JAG | 2017 | The structure of the COPI coat leaf | http://www.ebi.ac.uk/pdbe/entry/emdb/EMD-3720 | Publicly available at the EBI European Nucleotide Archive (accession no: EMD-3720) |
| Dodonova SO, Aderhold P, Kopp J, Ganeva I, Röhling S, Hagen WJH, Sinning I, Wieland F, Briggs JAG | 2017 | The structure of the COPI coat linkage I | http://www.ebi.ac.uk/pdbe/entry/emdb/EMD-3722 | Publicly available at the EBI European Nucleotide Archive (accession no: EMD-3722) |
| Dodonova SO, Aderhold P, Kopp J, Ganeva I, Röhling S, Hagen WJH, Sinning I, Wieland F, Briggs JAG | 2017 | The structure of the COPI coat linkage II | http://www.ebi.ac.uk/pdbe/entry/emdb/EMD-3723 | Publicly available at the EBI European Nucleotide Archive (accession no: EMD-3723) |
| Dodonova SO, Aderhold P, Kopp J, Ganeva I, Röhling S, Hagen WJH, Sinning I, Wieland F, Briggs JAG | 2017 | The structure of the COPI coat linkage IV | http://www.ebi.ac.uk/pdbe/entry/emdb/EMD-3724 | Publicly available at the EBI European Nucleotide Archive (accession no: EMD-3724) |
| Dodonova SO, Aderhold P, Kopp J, Ganeva I, Röhling S, Hagen WJH, Sinning I, Wieland F, Briggs JAG | 2017 | The structure of the COPI coat leaf | http://www.ebi.ac.uk/pdbe/entry/pdb/5NZR | Publicly available at the EBI European Nucleotide Archive (accession no: 5NZR) |
| Dodonova SO, Aderhold P, Kopp J, Ganeva I, Röhling S, Hagen WJH, Sinning I, Wieland F, Briggs JAG | 2017 | The structure of the COPI coat linkage I | http://www.ebi.ac.uk/pdbe/entry/pdb/5NZT | Publicly available at the EBI European Nucleotide Archive (accession no: 5NZT) |
| Dodonova SO, Aderhold P, Kopp J, Ganeva I, Röhling S, Hagen WJH, Sinning I, Wieland F, Briggs JAG | 2017 | The structure of the COPI coat linkage II | http://www.ebi.ac.uk/pdbe/entry/pdb/5NZU | Publicly available at the EBI European Nucleotide Archive (accession no: 5NZU) |

| | | | | |
|---|---|---|---|---|
| Dodonova SO, Aderhold P, Kopp J, Ganeva I, Röhling S, Hagen WJH, Sinning I, Wieland F, Briggs JAG | 2017 | The structure of the COPI coat linkage IV | http://www.ebi.ac.uk/pdbe/entry/pdb/5NZV | Publicly available at the EBI European Nucleotide Archive (accession no: 5NZV) |
| Dodonova S O, DiestelkoetterBachert P, von Appen A, Hagen W J, Beck R, Beck M, Wieland F, Briggs JAG | 2015 | The structure of the COPI coat triad | http://www.ebi.ac.uk/pdbe/entry/emdb/EMD-2985 | Publicly available at the EBI European Nucleotide Archive (accession no: EMD-2985) |
| Dodonova SO, DiestelkoetterBachert P, von Appen A, Hagen W J, Beck R, Beck M, Wieland F, Briggs JAG | 2015 | The structure of the COPI coat triad | http://www.ebi.ac.uk/pdbe/entry/pdb/5A1U | Publicly available at the EBI European Nucleotide Archive (accession no: 5A1U) |
| Faini M, Prinz S, Beck R, Schorb M, Riches JD, Bacia K, Brugger B, Wieland FT, Briggs JAG | 2015 | Structures from COPI-coated vesicles: triad | http://www.ebi.ac.uk/pdbe/entry/emdb/EMD-2084 | Publicly available at the EBI European Nucleotide Archive (accession no: EMD-2084) |

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
