## [Decision Letter]

Thank you for submitting your article "The structure of the COPI coat determined within the cell" for consideration by *eLife*. Your article has been reviewed by two peer reviewers, and the evaluation has been overseen by a Reviewing Editor and Randy Schekman as the Senior Editor. The following individuals involved in review of your submission have agreed to reveal their identity: Lauren P Jackson (Reviewer #2).

The reviewers have discussed the reviews with one another and the Reviewing Editor has drafted this decision to help you prepare a revised submission.

Summary:

Both reviewers found this work to be very exciting indeed, and the reviewing editor shares their enthusiasm. Reviewer 1 made the point that "this in-situ analysis of a coat is the dream of anyone working on the formation and delivery of such vesicles but also of anyone interested in subcellular compartmentalization and organelles." Reviewer 2 commented, "I share the authors' enthusiasm about how this will transform structural cell biology; this is really cool (pun intended)!" Both reviewers agreed that this study provides some major new mechanistic insights, in particular into cargo selection and vesicle uncoating. Their only other comments concern a few modifications they would like to see made to the text, either out of interest or for clarity, as listed below.

Minor points:

1) The reviewers would like some comment or speculation on why there does not appear to be cargo density below alpha-COP. In yeast, alpha-COP was long thought to be the most important subunit for recognizing KKxx motifs. Does *Chlamydomonas* lack KKxx cargoes and instead favor KxKxx motifs? Or does beta’-COP seem to prefer KKxx motifs here? This is an unexpected result, and it would be good to address it in light of published data, perhaps using bioinformatics tools to look at cargo motifs in *Chlamydomonas*. (In fact, one of the *Chlamydomonas* p24 family members, EDP07241, ends with KKxx, and these are normally quite abundant proteins.)

2) Figure 2—figure supplement 2: in parts D and E, it looks as though there is additional density next to the left-hand Arf1. Could this be an ArfGAP, as observed in the Dodonova paper? What ArfGAP(s) are found in *Chlamydomonas*? Would it make sense for it to be here? Since the related paper contains density for part of the ArfGAP, it would be good to discuss this observation.

3) Figure 2—figure supplement 3. The published structures modeled in densities corresponding to linkages I and II do not seem to fit well, in contrast to linkage IV. Can the authors comment on this poor fit with previously determined structures? Is it possible the in situ structure is different?

4) The phrasing 'in the vicinity of the di-lysine cargo binding site on b'-COP' might be clearer by saying: 'in the vicinity of the beta propellers of b'-COP, which contain a dilysine cargo-binding site.'

5) Introduction, small typo: “The coat-like subcomplex of COPI consists of epsilon, alpha, and beta' (prime) subunits…" (not beta).

---

## [Author Response]

Minor points:1) The reviewers would like some comment or speculation on why there does not appear to be cargo density below alpha-COP. In yeast, alpha-COP was long thought to be the most important subunit for recognizing KKxx motifs. Does Chlamydomonas lack KKxx cargoes and instead favor KxKxx motifs? Or does beta'-COP seem to prefer KKxx motifs here? This is an unexpected result, and it would be good to address it in light of published data, perhaps using bioinformatics tools to look at cargo motifs in Chlamydomonas. (In fact, one of the Chlamydomonas p24 family members, EDP07241, ends with KKxx, and these are normally quite abundant proteins.)

We have added the following text:

“…indicating that α-COP binds fewer, smaller, or more flexible cargos than β’–COP. […] In both cases, the difference in cargo binding could either be specific to *C. reinhardtii* or a widespread property.”

2) Figure 2—figure supplement 2: in parts D and E, it looks as though there is additional density next to the left-hand Arf1. Could this be an ArfGAP, as observed in the Dodonova paper? What ArfGAP(s) are found in Chlamydomonas? Would it make sense for it to be here? Since the related paper contains density for part of the ArfGAP, it would be good to discuss this observation.

The appearance of additional, unassigned density was an unfortunate artefact of the presentation. It is in fact ζ- COP. We adjusted the slice position by ~ 1 pixels and corrected the depth cueing on the PDB model to show that there is no additional density.

We did not observe any additional densities that could belong to ArfGAPs in the final average or in the averages derived specifically from buds and vesicles. We added a statement about this in the main text:“The lack of any extra densities in the averages from buds and vesicles further indicates that additional factors such as ArfGAPs bind only transiently or at low stoichiometry.”

3) Figure 2—figure supplement 3. The published structures modeled in densities corresponding to linkages I and II do not seem to fit well, in contrast to linkage IV. Can the authors comment on this poor fit with previously determined structures? Is it possible the in situ structure is different?

The linkages are determined at relatively low resolution in this work, and appear consistent with our previous studies at similar resolution (Faini et al. 2012).

In the case of linkage I, the structure is asymmetric (Dodonova et al. 2015), but we were unable to resolve the asymmetry at the low resolution of the current study. We had illustrated only one copy of alpha-epsilon in Figure 2—figure supplement 3, which was confusing when shown on top of a three-fold symmetrized density. We have now replaced it with three copies, and explained this more clearly in the figure legend (see below).

In the case of linkage II, one of the reasons for the poor fit the reviewers may refer to is that we had not displayed the peripheral gamma appendage in the fit. This has now been done. In the centre of the linkage the three-fold density into which the µ-homology domain fits is rather week. We do not believe this reflects a fundamental difference between the *in vitro* and *in vivo* systems, but it could reflect increased flexibility of the non-essential µ-homology domain *in vivo*. We have explained this more clearly in the figure legend: “Note that linkage I is an asymmetric structure, and the three copies of the complex between ε-COP and the C-terminal domain of α–COP may not all be in equivalent positions (Dodonova et al., 2015). […] ε-COP and the µ-homology domain of δ-COP are both dispensable for COPI function (Arakel et al., 2016).”

We had generated this figure using the deposited structures from Dodonova et al. 2015. We have generated the revised figure using the deposited structures from Dodonova et al. 2017, which are essentially the same, but at higher resolution.

4) The phrasing 'in the vicinity of the di-lysine cargo binding site on b'-COP' might be clearer by saying: 'in the vicinity of the beta propellers of b'-COP, which contain a dilysine cargo-binding site.'

Corrected (Results and Discussion, twelfth paragraph).

5) Introduction, small typo: “The coat-like subcomplex of COPI consists of epsilon, alpha, and beta' (prime) subunits…" (not beta).

Corrected (Introduction, second paragraph).

Additional changes:

In addition to the changes described above, we have also:

* Added EMDB accession numbers for all structures and representative tomograms.

* Added an additional supplementary video illustrating the segmented tomogram in Figure 1.

* Provided the Key Resources table and Transparent Reporting document.

* Added the following text to the Materials and methods section to further clarify the tomographic data collection: “Each tomogram was acquired from a separate cell, and thus each is a biological replicate. […] These tomograms were then sorted by image quality and tilt-series alignment precision, yielding 29 tomograms that were used for analysis…”